# Sleep-active neuron specification and sleep induction require FLP-11 neuropeptides to systemically induce sleep

Michal Turek[†], Judith Besseling[†], Jan-Philipp Spies, Sabine König, Henrik Bringmann*

Max Planck Institute for Biophysical Chemistry, Göttingen, Germany

**Abstract** Sleep is an essential behavioral state. It is induced by conserved sleep-active neurons that express GABA. However, little is known about how sleep neuron function is determined and how sleep neurons change physiology and behavior systemically. Here, we investigated sleep in *Caenorhabditis elegans,* which is induced by the single sleep-active neuron RIS. We found that the transcription factor LIM-6, which specifies GABAergic function, in parallel determines sleep neuron function through the expression of APTF-1, which specifies the expression of FLP-11 neuropeptides. Surprisingly FLP-11, and not GABA, is the major component that determines the sleep-promoting function of RIS. FLP-11 is constantly expressed in RIS. At sleep onset RIS depolarizes and releases FLP-11 to induce a systemic sleep state.

**\*For correspondence:** henrik. bringmann@mpibpc.mpg.de

[†]These authors contributed equally to this work

**Competing interests:** The authors declare that no competing interests exist.

## Introduction

Sleep is a systemic physiological state that is defined by behavioral criteria such as an increased arousal threshold, reversibility, and homeostatic regulation (*Campbell and Tobler, 1984*). Sleep behavior has been found in all animals that have a nervous system and that have been studied thoroughly (*Cirelli and Tononi, 2008*). Sleep homeostasis suggests that sleep is vital for animal life, and deprivation of this behavior typically has detrimental consequences (*Rechtschaffen and Bergmann, 1995*). Because of its importance, sleep is highly controlled by the nervous system, and failures of the sleep regulatory system cause disorders that are widespread in modern societies (*Panossian and Avidan, 2009*).

In mammals, wake and sleep are controlled by two antagonistic systems. The ascending system is wake-promoting (*Moruzzi and Magoun, 1949*; *Starzl et al., 1951*), and the descending system is sleep-promoting (*von Economo, 1930*; *Nauta, 1946*; *McGinty and Sterman, 1968*; *McGinty et al., 2004*; *Saper et al., 2005*). Central to the control of sleep are sleep-active sleep-promoting neurons such as those located in the preoptic area (Ventral Lateral Preoptic Nucleus, VLPO and Median Preoptic Nucleus, MnPO), the parafacial zone of the medulla, and the thalamic reticular nucleus (*McGinty et al., 2004*; *Anaclet et al., 2012*; *2014*; *Alam et al., 2014*; *Ni et al., 2016*). These neurons typically fire preferentially at the onset of sleep, they actively induce sleep, and they express the neurotransmitter GABA. The VLPO also expresses the neuropeptide Galanin (*Sherin et al., 1998*; *Szymusiak et al., 1998*; *Gaus et al., 2002*). Interestingly, small brain areas can induce the global state of sleep that affects all areas of the brain and also other organs. It has been proposed that sleep neurons induce sleep through projections to arousal centers (*Saper et al., 2005*; *Sherin et al., 1996*). To ensure that sleep and wake are manifested as discrete states, the ascending

**eLife digest** Sleep keeps us healthy and happy, and is essential for all animals. Specialized neurons in the brain become highly active to generate this restful state. There are relatively few of these "sleep-active" neurons in the brain, but they are able to control sleep in the entire animal.

Like most other neurons, sleep-active neurons release substances called neurotransmitters. The sleep-active neurons in many different species release a neurotransmitter called GABA, although they also contain other neurotransmitters called neuropeptides that were thought to be less important for triggering sleep.

The roundworm *Caenorhabditis elegans* has become an important model system for studying the molecular biology of sleep as it contains only one sleep-active neuron. Turek et al. have now studied this *C. elegans* neuron and have discovered transcription factors – proteins that control gene expression – that are required for the sleep-active neuron to induce sleep.

Further investigation revealed that the transcription factors specify the production of a neuropeptide called FLP-11. The sleep-active neuron always contains FLP-11, but only releases it as sleep begins. Once released, FLP-11 moves onto target cells to induce sleep in the entire organism. Thus, FLP-11 – and not GABA – is the major sleep-inducing neurotransmitter in *C. elegans*.

To induce a sleep state throughout an entire organism, the activities of many different cells must be controlled. A future challenge will be to figure out how FLP-11 does this.

and descending systems mutually inhibit each other in a flip-flop switch (*Gallopin et al., 2000*; *Saper et al., 2001*).

*Caenorhabditis elegans* has become an invaluable model system for molecular dissection of biological processes (*Brenner, 1974*). It is amenable to genetics, has a small and invariant nervous system of just 302 neurons, and it is transparent (*Brenner, 1974*; *White et al., 1986*; *Chalfie et al., 1994*). In *C. elegans*, quiescence behavior can be found in satiated adults, after stress, during dauer diapause, and during larval development (*Cassada and Russell, 1975*; *You et al., 2008*; *Hill et al., 2014*). For some of these types of quiescence, it is yet unclear how they relate to sleep and they are, at least in part, regulated by different mechanisms (*Trojanowski et al., 2015*).

Here, we focus on a well-characterized developmentally controlled sleep behavior that can be found in *C. elegans* larvae prior to each of the four molts (*Cassada and Russell, 1975*). Developmentally controlled sleep fulfills the criteria that define sleep in other organisms (*Raizen et al., 2008*; *Trojanowski and Raizen, 2016*). These criteria are reversibility, an increased arousal threshold, and homeostatic regulation (*Raizen et al., 2008*; *Jeon et al., 1999*; *Schwarz et al., 2011*; *Driver et al., 2013*; *Iwanir et al., 2013*; *Nagy et al., 2014*). Further analysis has shown that sleep behavior in *C. elegans* and sleep in other organisms are controlled by homologous genes such as *period/lin-42*, Notch signaling, EGF signaling and several other molecules including neurotransmitter systems (*Nagy et al., 2014*; *Monsalve et al., 2011*; *Van Buskirk and Sternberg, 2007*; *Singh et al., 2011*; *Singh et al., 2014*; *Choi et al., 2013*). These molecular similarities suggest that sleep behavior in *C. elegans* and sleep in other organisms share a common evolutionary origin.

Sleep behavior in *C. elegans* has been shown to profoundly change the activity of neurons and muscles (*Schwarz et al., 2011*; *Iwanir et al., 2013*; *Cho and Sternberg, 2014*; *Schwarz et al., 2012*). It requires the activity of the single interneuron RIS (neuron class of one ring interneuron; *White et al., 1986*). This neuron is active at the onset of sleep, it actively induces sleep, and it expresses GABA (*Turek et al., 2013*). Thus, RIS is similar to sleep-active neurons in mammals.

In order to be sleep-inducing, RIS requires APTF-1, a highly conserved transcription factor of the AP2 family. Without APTF-1, RIS is still sleep-active but can no longer induce sleep (*Turek et al., 2013*). In humans, mutation in the AP2 homolog TFAP2beta causes Char syndrome, which is linked to insomnia or sleepwalking (*Mani et al., 2005*). Together, this supports the view that sleep-neurons and AP2 transcription factors are conserved regulators of sleep. However, the mechanism of how APTF-1 renders RIS sleep promoting is unclear.

Here, we identify a gene regulatory system that determines the sleep-inducing function of RIS. In this network, a transcription factor that controls GABAergic function in a subset of neurons, LIM-6, in

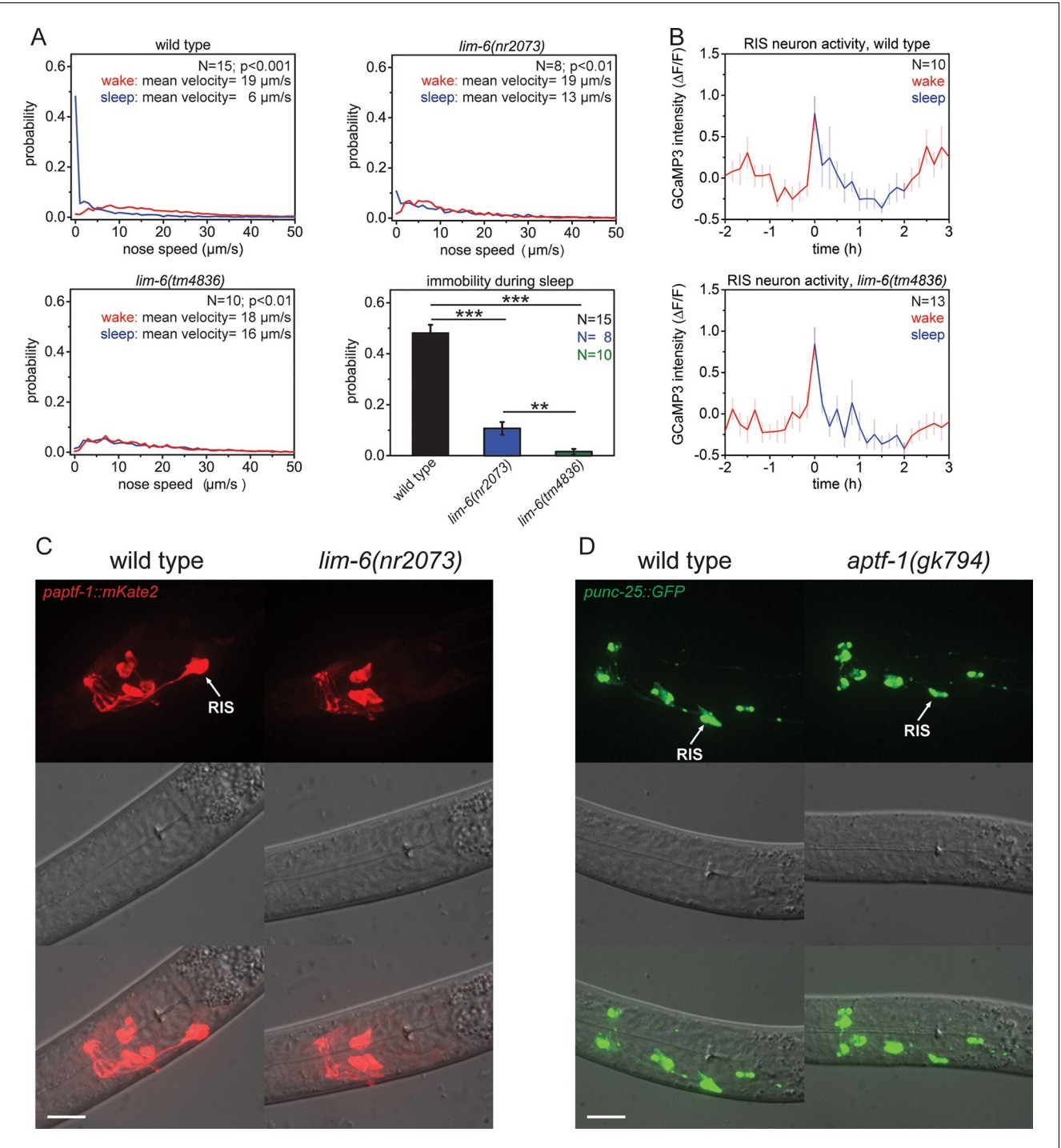

**Figure 1.** The LIM homeobox transcription factor LIM-6 controls sleep by specifying expression of the transcription factor APTF-1 in RIS. (**A**) Probability distribution of nose speeds during wake and sleep for wild type and *lim-6* mutants. *lim-6(nr2073)* shows substantially reduced and *lim-6(tm4836)* shows a complete lack of immobility during the time the animals should be sleeping. (**B**) Averaged RIS calcium activity pattern across time in wild type and *lim-6(tm4836)*. RIS is active at the onset of sleep in wild type and in *lim-6(tm4836)*. There was no statistically significant difference between wild-type and *lim-6* worms (p > 0.05, Welch test). (**C**) Expression of *paptf-1::mKate2* in wild type and *lim-6(nr2073)* L1 larvae. Expression of mKate2 is absent in RIS in *lim-6(nr2073)* showing that LIM-6 controls expression of APTF-1 in RIS. (**D**) Expression of *punc-25::GFP* in wild type and *aptf-1(gk794)*. Reporter GFP expression is normal in RIS in *aptf-1* mutant worms indicating that GABAergic function is not controlled by APTF-1. Statistical tests used were Wilcoxon Signed Paired Ranks test for comparison within the same genotype and Student's t-test for comparisons between genotypes. Error bars are SEM. ** denotes statistical significance with p<0.01, *** denotes statistical significance with p<0.001. Scale bars are 10 μm.

*Figure 1 continued on next page*

*Figure 1 continued*

The following figure supplement is available for figure 1:

**Figure supplement 1.** LIM-6 controls expression of APTF-1 across development but does not control the expression of the GABA vesicular transporter gene *unc-47*.

parallel controls the expression of the APTF-1 transcription factor. APTF-1, in turn, specifies the expression of sleep-inducing FLP-11 peptides. FLP-11 is always present in RIS, and thus, this neuron can induce sleep at any time it gets activated. At sleep onset, calcium transient activity of RIS increases and leads to the release of FLP-11 peptides, which induce quiescence. Thus, we show that sleep can be induced systemically by the single RIS neuron through FLP-11 release.

## Results

### LIM-6 controls APTF-1 in RIS

Sleep-active neurons express GABA in both mammals and *C. elegans*. The LIM homeobox transcription factor LIM-6 is expressed in a subset of GABAergic neurons including RIS and has been shown to be required for some aspects of GABAergic neuron specification, including the expression of the GABA-synthesizing enzyme glutamate decarboxylase UNC-25 (*Hobert et al., 1999*; *Jin et al., 1999*). Thus, we tested whether LIM-6 is involved in sleep control. First, we investigated the spontaneous behavior of two *lim-6* mutants, which contain large deletions and represent strong loss-of-function mutations. We cultured worms in microfluidic compartments, filmed their activity over the sleep-wake cycle, and quantified their locomotion behavior by tracking nose movement. Similar to the wild type, *lim-6* mutants stopped feeding before the molt, allowing the identification of lethargus, the developmental time the larvae should be sleeping. The mutants had strongly reduced or even complete absence of immobility during the non-pumping phase (*Figure 1A*). We then quantified RIS activation in *lim-6* mutant worms by imaging the calcium indicator GCaMP3 expressed in RIS (*Tian et al., 2009*; *Butler, 2012*). We found that RIS activity increased at sleep onset in wild-type animals and also normally increased in *lim-6* mutants during the time the animal should enter sleep (*Figure 1B*). Because the sleep phenotype of the *lim-6* mutant worms was similar to the sleep phenotype of *aptf-1* mutants (*Turek et al., 2013*), we tested whether *lim-6* controls *aptf-1* expression. We crossed a line expressing mKate2, a red-fluorescent protein (*Shcherbo et al., 2009*), under the control of the *aptf-1* promoter into *lim-6* mutant worms and quantified the expression of mKate2 in RIS. *aptf-1* expression was completely abolished in most individuals and strongly reduced in the remaining ones in all developmental stages (*Figure 1C*, *Figure 1—figure supplement 1A*). We also looked at GABAergic function in *aptf-1* mutant worms and found that *aptf-1* did not control the expression of *unc-25* or the vesicular GABA transporter gene *unc-47* (*Figure 1D*, *Figure 1—figure supplement 1B*). Thus, LIM-6 does not appear to affect sleep primarily through GABAergic function determination. Rather, LIM-6 controls sleep-promoting function primarily through a parallel pathway that depends on APTF-1.

### APTF-1 controls gene expression in RIS

How is sleep-promoting function generated by APTF-1? AP2 transcription factors are highly conserved regulators of gene expression (*Zhao et al., 2011*; *Eckert et al., 2005*) suggesting that *aptf-1* acts through gene expression in RIS. Thus, we determined the transcriptional profile of *aptf-1* mutants. We performed transcriptional profiles of *aptf-1* mutants using microarrays using two conditions. First, because *aptf-1* is most strongly expressed during late embryogenesis, we analyzed pretzel-stage embryos. Second, because *aptf-1* is required for sleep behavior, we analyzed sleeping L4 larvae. The microarray experiment revealed that *flp-11*, a neuropeptide gene, was strongly downregulated in *aptf-1* mutant worms in both embryos and larvae (*Figure 2A*, *Supplementary file 1*, Table 1A and 1B). We thus further analyzed the role of *flp-11* in sleep control. We also analyzed some other genes that were regulated by *aptf-1*, but none of them appeared to be important for sleep regulation (*Figure 2—figure supplements 1* and *2*). To test whether *flp-11* is expressed in RIS and to verify its regulation by *aptf-1*, we generated an mKate2 promoter fusion as a reporter line.

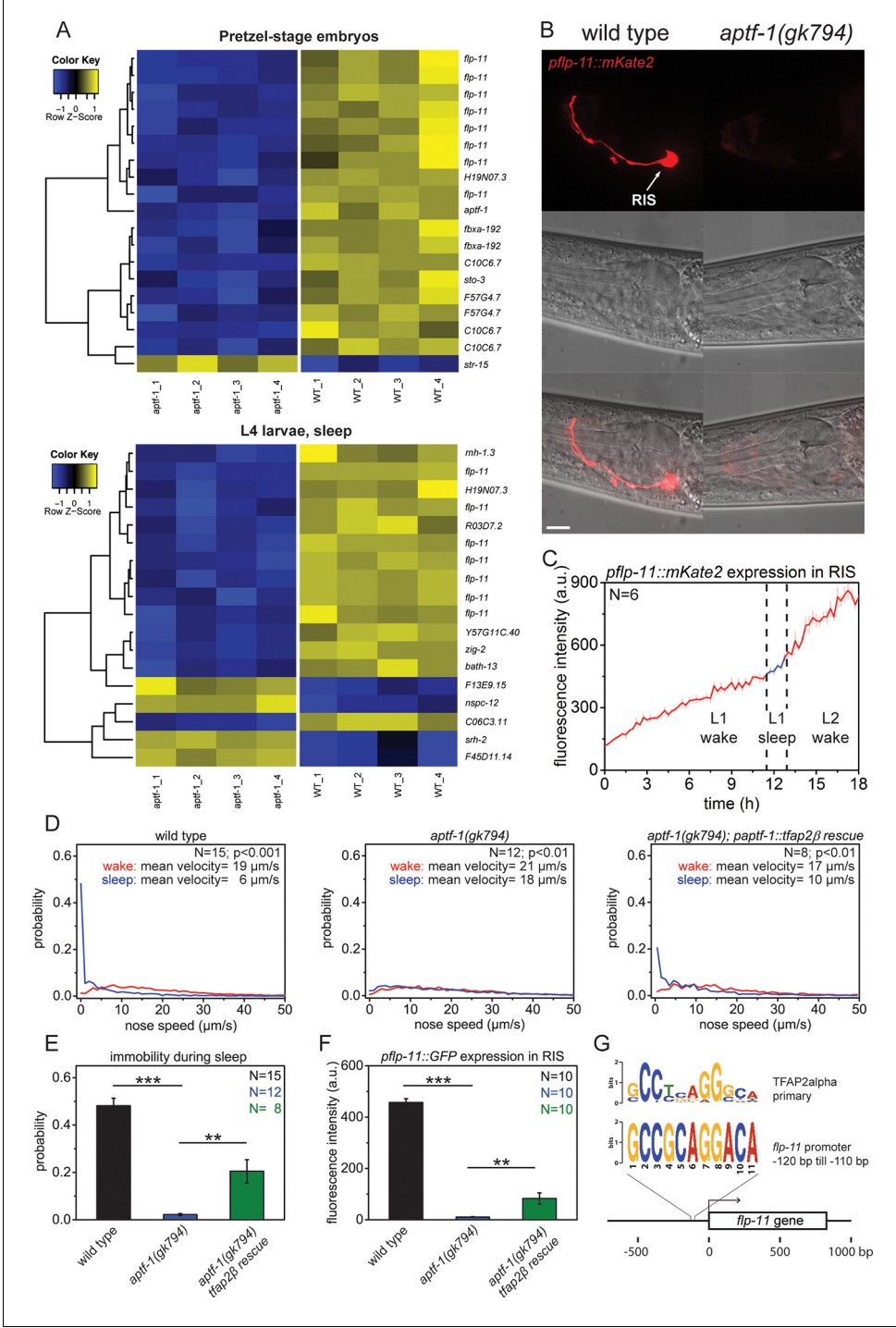

**Figure 2.** The AP2 transcription factor APTF-1 controls FLP-11 expression in RIS. (**A**) Transcriptional analysis of *aptf-1(gk794)* mutants revealed genes that are regulated by APTF-1. Wild-type and *aptf-1(gk794)* pretzel-stage embryos and sleeping L4 larvae were used for a transcriptome analysis. In both life stages, expression of the FMRFamide-like neuropeptide FLP-11 was strongly reduced in *aptf-1(gk794)*. This suggests transcriptional control of FLP-11 by APTF-1. Data can be found in ***Supplementary file 1***, Tables 1A and 1B. (**B**) Expression of *pflp-11:: mKate2* in wild type and *aptf-1(gk794)*. Expression of mKate2 was absent in RIS in *aptf-1(gk794)* showing that APTF-1 controls expression of FLP-11. Expression of *flp-11* in RIS was reminiscent to the expression of the *flp-11* homolog *afp-6* in RIS in *Ascaris* nematodes (***Yew et al., 2007***). Expression for additional genes can be found in ***Figure 2—figure supplements 2*** and ***3***. (**C**) *flp-11* expression profile in RIS over the sleep-wake cycle. Expression does not change with the sleep-wake cycle. (**D**) Probability distribution of nose speeds during wake and sleep for

*Figure 2 continued on next page*

*Figure 2 continued*

wild type, *aptf-1(gk794)* and *aptf-1(gk794); paptf-1::tfap2β* rescue. (E) Comparison of immobility during sleep for wild type, *aptf-1(gk794)*, and *aptf-1(gk794); paptf-1::tfap2β*. The mouse TFAP2β partially rescued the *aptf-1(gk794)* sleep phenotype. (F) Comparison of *pflp-11::GFP* fluorescence intensity in RIS for wild type, *aptf-1(gk794)*, and *aptf-1(gk794); paptf-1::tfap2β*. The mouse TFAP2β partially rescued the expression of *flp-11* in RIS (18% of wild-type level). (G) Analysis of putative AP2-binding sites in the *flp-11* promoter region. The *flp-11* promoter region was scanned for the primary mouse AP2α-binding site. Overlap was found (p<0.001, q=0.06 (Grant et al., 2011)) for one binding site. Statistical test used was Wilcoxon Signed Paired Ranks test. ** denotes statistical significance with p<0.01, *** denotes statistical significance with p<0.001. Scale bar is 10 μm.

The following figure supplements are available for figure 2:

**Figure supplement 1.** C10C6.7 is a putative four transmembrane helix protein that is expressed in RIS and that is controlled by *aptf-1*.

**Figure supplement 2.** Expression pattern of *sto-3* and *H19N07.3*.

**Figure supplement 3.** FLP-11 is strongly expressed in RIS and weakly in additional neurons. APTF-1 controls the expression in RIS.

**Figure supplement 4.** Mouse TFAP2beta partially restores expression of *flp-11* neuropeptides in RIS in *aptf-1* mutant worms.

We then checked the expression in wild-type and *aptf-1* mutant backgrounds using fluorescence microscopy. *flp-11* was expressed strongly in RIS and faintly in a few additional neurons. Expression was abolished or greatly reduced in RIS in *aptf-1* mutant worms (*Figure 2B*, *Figure 2—figure supplement 3*). We followed the expression of *flp-11:mKate2* over the sleep-wake rhythm and found that mKate2 was constantly expressed during larval development both during sleep and wake (*Figure 2C*). Transcription factors are often structurally conserved, and the homolog from a highly divergent species can replace the mutation of the endogenous factor (*Halder et al., 1995*). There are five homologs of APTF-1 in mammals, designated TFAP2-alpha to TFAP2-epsilon (*Eckert et al., 2005*). We tested for structural conservation of AP2 by expressing the mouse homolog of *aptf-1*, *tfap2beta*, in RIS using the *aptf-1* promoter. For our experiment, we chose TFAP2beta, because it has been linked to insomnia in humans (*Mani et al., 2005*). Nose speed measurements during sleep showed that *tfap2beta* expression partially restored immobility (*Figure 2D,E*). Expression of mouse *tfap2beta* also partially restored the expression of *flp-11* in *aptf-1* mutant worms (*Figure 2F*, *Figure 2—figure supplement 4*). The structural conservation of AP2 transcription factors suggests that also the DNA binding site is conserved. In fact, it has been shown that the binding site of AP2 is conserved over 600 million years of bilaterian evolution, and the different mammalian AP2 paralogs have nearly identical binding sites (*Nitta et al., 2015*). Thus, we searched for mammalian AP2 transcription factors binding sites in the promoter region of *flp-11* (*Eckert et al., 2005*; *Grant et al., 2011*). Indeed, we found a putative AP2-binding site in the *flp-11* promoter region, consistent with the regulation of *flp-11* by *aptf-1* (*Figure 2G*). These results show that *aptf-1* is required for *flp-11* expression in RIS.

## RIS induces sleep through FLP-11

We next tested whether *flp-11* is required for sleep behavior. We analyzed an *flp-11* deletion that is predicted to affect all four peptides encoded by the gene and filmed and quantified sleep behavior as before. *flp-11* mutant larvae showed a normal pre-molting cessation of pumping. However, locomotion quiescence was strongly reduced in *flp-11* mutant larvae. Although nose immobility was 48% in wild type, it was strongly reduced to less than 14% in *flp-11* mutants (*Figure 3A*). To verify that the sleep phenotype observed in *flp-11* mutants was caused by *flp-11* deletion, we tested whether a wild-type copy of *flp-11* would restore locomotion quiescence during sleep. We found that the wild-type transgene could rescue the *flp-11* mutant phenotype (*Figure 3A*). If *flp-11* is a major target of *aptf-1*, a wild-type copy of *flp-11* should also rescue, at least partially, the sleep phenotype of *aptf-1*

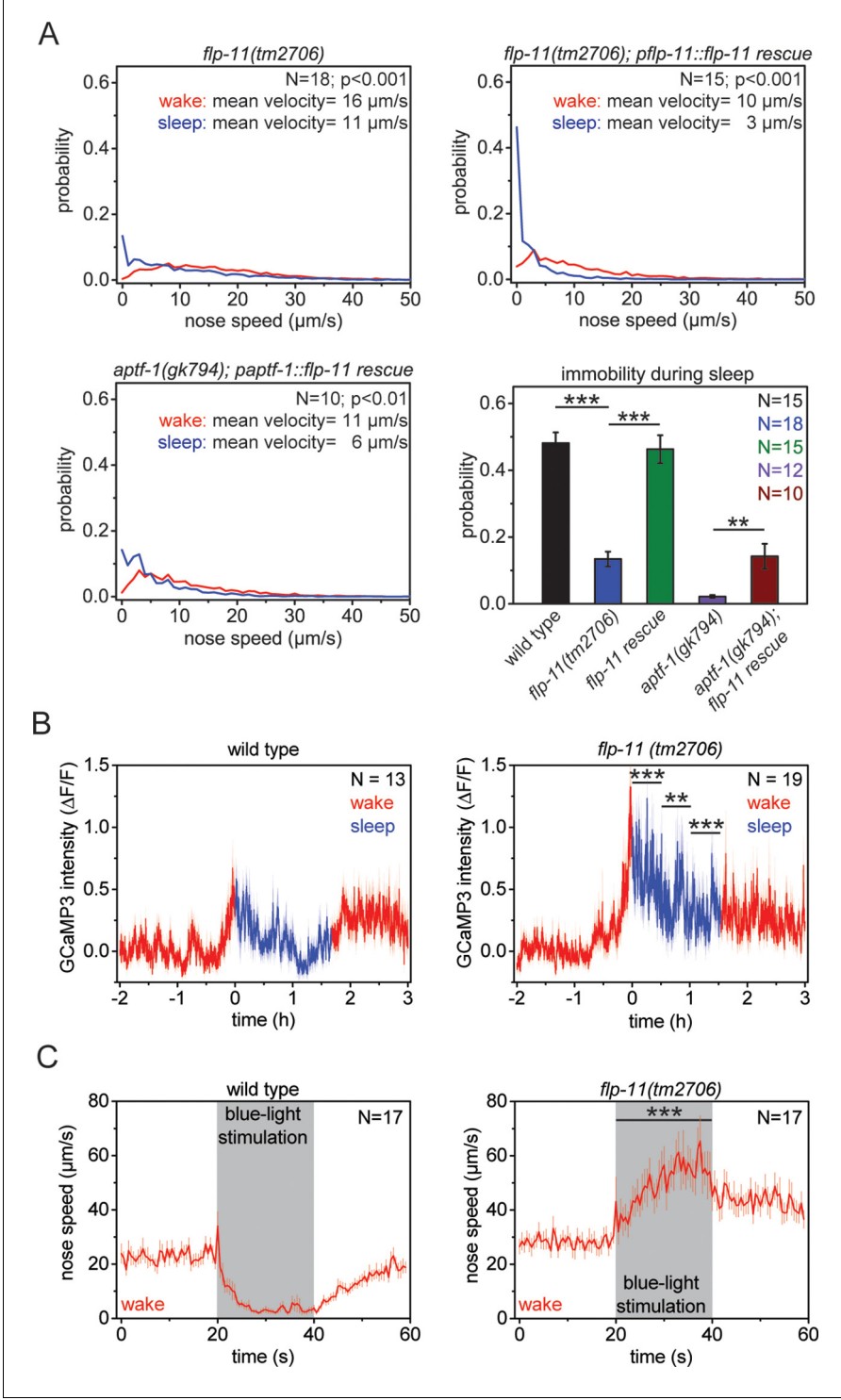

**Figure 3.** RIS induces sleep through the sleep-inducing FMRFamide-like neuropeptide FLP-11. (**A**) Probability distribution of nose speeds during wake and sleep for wild type, *flp-11(tm2706)*, *flp-11(tm2706); pflp-11::flp-11* rescue and *aptf-1(gk794); paptf-1::flp-11* rescue. Immobility during the time the animal should be sleeping was substantially reduced in *flp-11(tm2706)*. *flp-11(tm2706)* could be rescued by expression of the wild-type *flp-11* gene. Furthermore, expression of *flp-11* in *aptf-1(gk794)* partially rescued sleep behavior. (**B**) Averaged RIS calcium activity pattern across time in wild type and *flp-11(tm2706)*. RIS was strongly activated at the onset of sleep in *flp-11(tm2706)* (Student's t-test). (**C**) Channelrhodopsin-2 activation of *aptf-1*-expressing neurons caused immediate immobility in wild type. In contrast, *flp-11(tm2706)* accelerated upon blue light stimulation showing that RIS-
*Figure 3 continued on next page*

*Figure 3 continued*

dependent immobility is impaired. Statistical tests used were Wilcoxon Signed Paired Ranks test for comparisons within genotypes and Student's t-test for comparisons between genotypes. Error bars are SEM. ** denotes statistical significance with p<0.01, *** denotes statistical significance with p<0.001.

mutants. Because the *flp-11* promoter is regulated by *aptf-1*, we used the *aptf-1* promoter to drive expression of *flp-11* in *aptf-1* mutants. Whereas *aptf-1* mutants without the transgene did not show any detectable immobility, the *flp-11* transgene partially restored immobility (*Figure 3A*). It is likely that *aptf-1* acts through additional targets, which may explain why the rescue observed after *flp-11* expression in the *aptf-1* mutant was partial and small. We next investigated the activation of RIS at sleep onset in *flp-11* mutants using GCaMP3. RIS strongly activated at the onset of the non-pumping period in *flp-11* mutant worms (*Figure 3B*). Thus, *flp-11* is not required for activation of RIS at sleep onset. To test whether sleep induction by RIS is impaired in *flp-11* mutant worms, we optogenetically activated RIS with Channelrhodopsin-2 during wake and followed the behavioral response of the worms by nose tracking (*Nagel et al., 2005*). Although wild-type animals showed a reduction in movement and became immobile after blue light illumination, *flp-11* mutant worms did not decrease their movement but rather increased it (*Figure 3C*). If FLP-11 peptides are sleep-promoting, then ectopic overexpression during wake may induce anachronistic quiescence (*Singh et al., 2011*; *Nelson et al., 2013*; *2014*). We overexpressed FLP-11 in adult worms using a heat-shock-inducible promoter that drives broad expression in the nervous system and other tissues (*Jones et al., 1986*). After a 5-min heat shock, we followed the fraction of immobilized worms over time. Adult worms that were expressing FLP-11 driven by the heat shock promoter became immobile 1–2 hr after the heat shock. Control worms that were heat shocked but did not express the transgene or expressed other *flp* genes did not show any immobilization (*Figure 4A*, *Figure 4—figure supplement 1*). Taken together these data imply that RIS induces sleep through FLP-11. These peptides are expressed in RIS during both sleep and wake, and optogenetic activation of RIS can induce quiescence during wake. This suggests a model in which RIS can induce sleep at any time. According to this model, RIS depolarizes at sleep onset and it releases FLP-11, which then induces sleep.

## G-protein-coupled receptors in sleep

Many neuropeptides act through G-protein-coupled receptors (*Peymen et al., 2014*; *Frooninckx et al., 2012*). To search for effectors through which FLP-11 induces sleep, we investigated three neuropeptide receptors that are activated by FLP-11 peptides in *in vitro* assays. These receptors are FRPR-3, NPR-4, and NPR-22 (*Frooninckx et al., 2012*; *Mertens et al., 2006*; *Mertens et al., 2004*; *Cohen et al., 2009*). We first tested whether deletion of these receptors can suppress FLP-11-induced anachronistic quiescence. We crossed receptor deletions into our FLP-11 overexpressing line and quantified heat-shock-induced quiescence. We found that the maximum quiescence was reduced in each of these mutants, albeit only slightly (*Figure 4—figure supplement 2A*). Thus, we tested whether these receptors act redundantly by testing all double mutant permutations and a triple mutant containing all three receptor deletions at the same time. The double mutants had further decreased quiescence, and the triple mutant had the strongest reduction in quiescence (*Figure 4A*, *Figure 4—figure supplement 2A*). Thus, the quiescence induced by FLP-11 overexpression partly depends on multiple effectors. We next investigated sleep behavior in the receptor mutants. Whereas the single receptor mutants did not show a significant reduction in immobility, the receptor triple mutant showed a small increase in nose speed during sleep and a small reduction of immobility (*Figure 4B*, *Figure 4—figure supplement 2B*). We investigated transgenic animals expressing promoter fusions containing the putative FLP-11 receptor and mKate2. To investigate the subcellular localization, we made transgenic animals expressing GFP-tagged proteins for the three receptors (*Sarov et al., 2012*). The promoter fusions showed that FRPR-3 expressed in approximately 30 neurons, mostly in the head. NPR-4 expressed in five neurons (*Cohen et al., 2009*). NPR-22 expressed in several neurons, muscle tissue in the pharynx and in the head (*Figure 4C*). Interestingly, the receptors were expressed in neurons that are not postsynaptic to RIS (*White et al., 1986*). The only exception that we found was the AVK neuron, which is postsynaptic

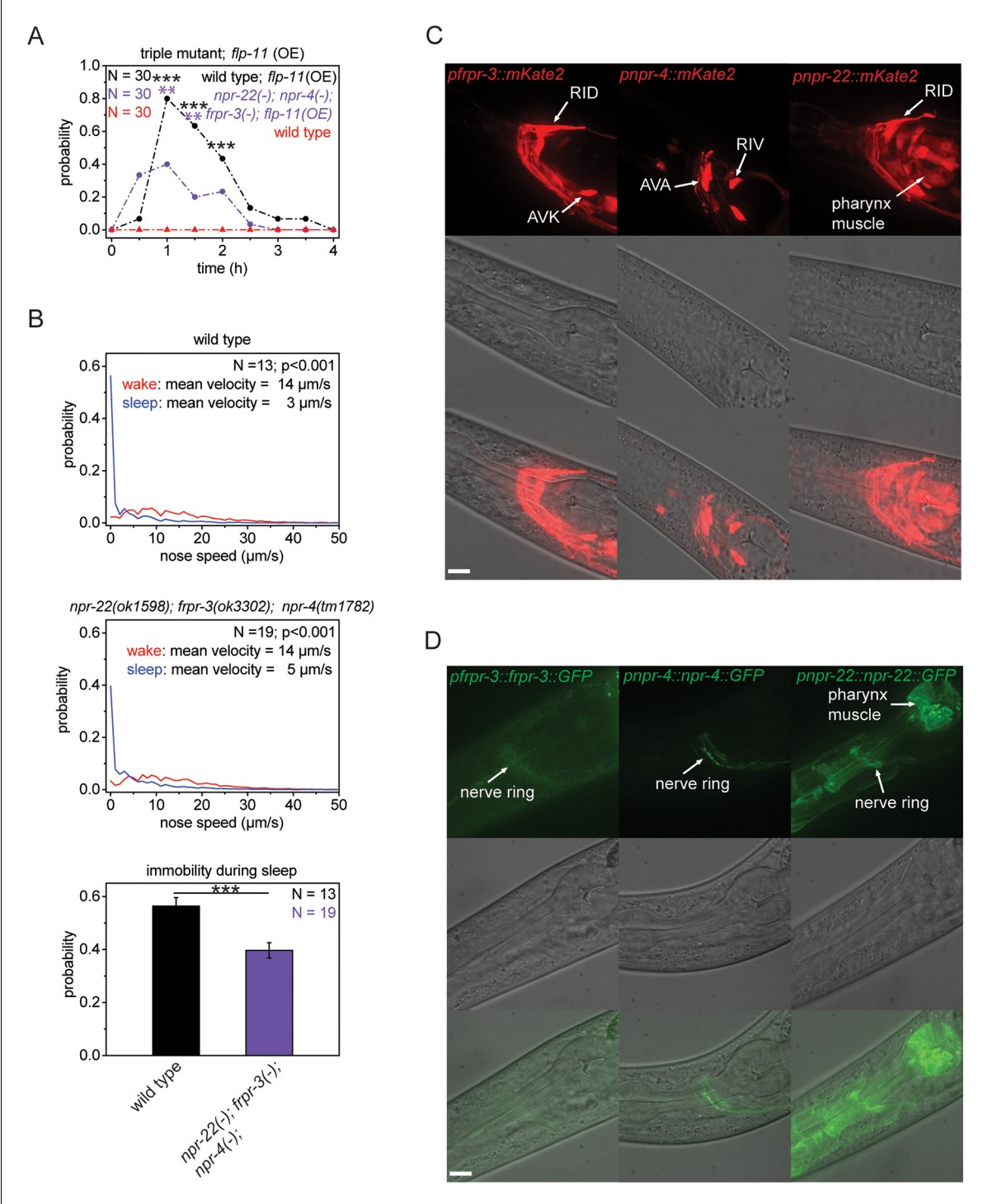

**Figure 4.** Multiple receptors may be involved in sleep induction. (**A**) Behavioral analysis over time after the heat-shock-induced overexpression of *flp-11* in wild-type and the *npr-22(ok1598); frpr-3(ok3302); npr-4(tm1782)* triple mutant. To assess the effect of the heat shock on quiescence, wild-type worms without *flp-11* overexpression were analyzed at the same time and do not show any behavioral changes. Overexpression of *flp-11* caused anachronistic quiescence that was lasting approximately 1 hr in the wild type. Quiescence was significantly reduced by approximately 50% in the triple mutant. (**B**) Probability distribution of nose speeds during wake and sleep for wild type and *npr-22(ok1598); frpr-3(ok3302); npr-4(tm1782)* triple mutant. Immobility during sleep was reduced by about 30% in the *npr-22(ok1598); frpr-3(ok3302); npr-4(tm1782)* triple mutant. (**C**) Expression patterns of *frpr-3, npr-4* and *npr-22* promoter fusions. FRPR-3 is expressed in about 30 neurons, mostly in the head. Expression of NPR-4 was seen in about five neurons. NPR-22 was expressed in several neurons and muscle tissue including pharynx and head muscle. (**D**) Expression patterns of GFP-tagged fosmids for *frpr-3, npr-4*, and *npr-22*. FRPR-3 and NPR-4 were mostly expressed around the nerve ring. NPR-22 localized broadly to the plasma membrane in several neurons,

*Figure 4 continued on next page*

*Figure 4 continued*

pharynx muscle, head muscle, and the anal sphincter muscle. Statistical tests used were Wilcoxon Signed Paired Ranks test for comparisons within genotypes and Student's t-test for comparisons between genotypes. Error bars are SEM. ** denotes statistical significance with p<0.01, *** denotes statistical significance with p<0.001. Scale bars are 10 μm.

The following figure supplements are available for figure 4:

**Figure supplement 1.** Heat-shock-induced *flp-11* overexpression causes quiescence but heat-shock-induced overexpression of three other *flp* genes does not, suggesting that quiescence cannot be induced by overexpression of any *flp*.

**Figure supplement 2.** Single receptors mutants do not show reduced quiescence during sleep, but do show reduced quiescence upon heat-shock-induced overexpression of *flp-11*.

to RIS and expressed FRPR-3. Translational fusions showed that FRPR-3 and NPR-4 were weakly expressed and localized mostly to the nerve ring, whereas NPR-22 was localized broadly to the plasma membrane of neurons, and muscles of the pharynx, head, and muscle (*Figure 4D*). The phenotypes for the GPCRs that we observed were small and the binding affinities of FLP-11 peptides to these GPCRs in vitro were low which makes it premature to conclude that these receptors act by binding of FLP-11 (*Frooninckx et al., 2012*; *Mertens et al., 2004*; *2006*; *Cohen et al., 2009*). It could be that FLP-11 acts through one main receptor, which remains unidentified. Alternatively, FLP-11 could act through several redundant receptors, which may include FRPR-3, NPR-4, and NPR-22.

## Discussion

### The role of GABA in sleep induction by RIS

GABA has been proposed to play a major role in sleep function. Its conserved expression in sleep-active neurons suggests that it has an important function in these neurons. Enhancers of GABAergic neurotransmission have been used to treat sleeping problems and GABA has been suggested to play a role in sleep induction in several systems including *C. elegans* (*Singh et al., 2014*; *Dabbish and Raizen, 2011*). In contrast to the common view that GABA is the major sleep inducer in sleep-active neurons, we did not find evidence that GABA is the major sleep-inducing transmitter in RIS. This is consistent with our previous observation that optogenetic activation of RIS still causes quiescence in *unc-25* mutant worms and that *unc-25* mutant worms still show sleeping behavior (*Turek et al., 2013*). In addition, here we show that GABAergic function induction can be separated from sleep neuron function downstream of *lim-6*. This suggests that GABA plays a rather minor role in sleep-induction in RIS and that we did not detect it. More specific and more sensitive assays may resolve the question of the role of GABA in RIS in the future.

### FLP-11 is the major sleep-promoting transmitter of RIS

Our results show that FLP-11 is a crucial sleep-inducing component in RIS and an important target of *aptf-1*. In mammals, sleep neurons of the VLPO express the inhibitory neuropeptide Galanin, and projections extend to the tuberomammillary nucleus, which expresses Galanin receptors (*Sherin et al., 1998*; *Gaus et al., 2002*). The locus coeruleus, a wake-promoting brain region, is also innervated by VLPO projections and can be inhibited by Galanin administration (*Seutin et al., 1989*; *Pieribone et al., 1995*). Also, Galanin has been shown to have sedating effects on both zebrafish and human subjects (*Woods et al., 2014*; *Murck et al., 2004*). However, sleep phenotypes for Galanin knockouts have not been reported, despite being available for several years (*Wynick et al., 1998*; *Kerr et al., 2000*). These experiments suggest that Galanin has a modulatory role on sleep, but may not be central to sleep induction. Galanin does not appear to be homologous to FLP-11, as it belongs to the family of Galanin peptides (*Lang et al., 2007*), whereas *flp-11* encodes peptides of the RFamide family (*Li et al., 1999*). Also, unlike FLP-11, Galanin is expressed widely in the brain and has diverse functions (*Maria Vrontakis, 2002*). In *Drosophila*, sleep requires a neuropeptide called sNPF, which may be functionally similar to FLP-11 as both are inhibitory and are released from

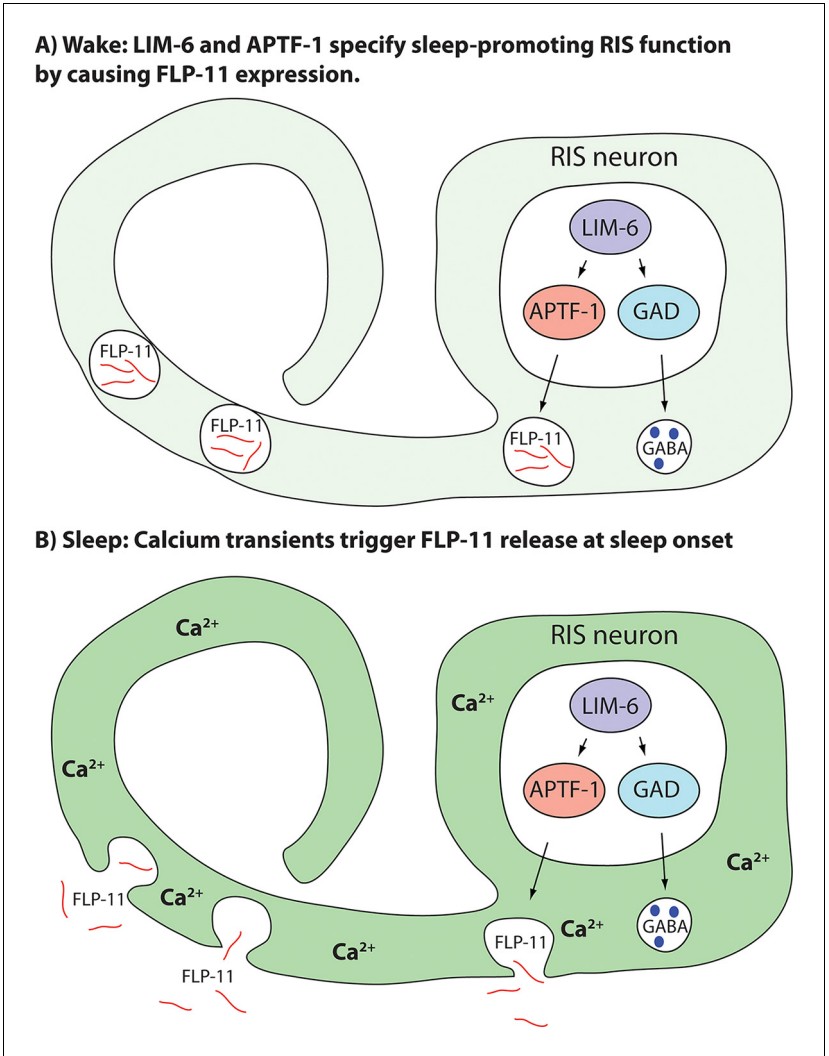

**Figure 5.** Model for generation of sleep-promoting function of RIS and sleep induction by RIS. According to this model, the transcription factor LIM-6 controls GABAergic and peptidergic function in RIS in parallel. To render this neuron sleep-promoting, LIM-6 is required for the expression of the APTF-1 transcription factor. APTF-1, in turn, is required for the expression of sleep-inducing FLP-11 peptides. FLP-11 is present in RIS at all times. Sleep onset is triggered by an unknown signal, which leads to a depolarization and to calcium influx. This triggers FLP-11 release, which in turn systemically induces sleep behavior.

sleep-promoting neurons (*Shang et al., 2013*; *Vecsey et al., 2014*; *Chen et al., 2013*). Thus, inhibitory neuropeptides appear to play important roles in sleep-promoting neurons across species.

## A model for sleep neuron specification and sleep induction through RIS

Taken together, we present a model for how sleep-promoting function is generated and for how sleep is induced (*Figure 5*). In this model, the transcription factor LIM-6 separately controls GABAergic and sleep-promoting functions. Sleep-promoting function is mediated by the expression of the APTF-1 transcription factor, which is crucially required for sleep induction by RIS. APTF-1, in turn, is required for the expression of sleep-inducing FLP-11 peptides. FLP-11 is always present in RIS allowing the induction of sleep at any time the neuron activates. At sleep onset, an unknown signal triggers depolarization and calcium influx in RIS, which then triggers release of FLP-11 peptides to systemically induce sleep behavior.

## Materials and methods

### Worm maintenance and strains

*C. elegans* worms were grown on Nematode Growth Medium (NGM) plates seeded with *E. coli* OP50 at 25°C as described (*Brenner, 1974*). The strains and alleles that were used in this study can be found in *Supplementary file 2*. The deletion alleles were backcrossed two to ten times against N2 to generate HBR lines (the exact number of backcrosses for each strain is indicated in brackets). Insertions were backcrossed two times against N2 to remove the *unc-119(-)* background. Back-crossed strains were the basis for all experiments. During backcrossing, the genotypes were followed by PCR. Primers to detect the deletions using a three primer PCR can be found in *Supplementary file 3*.

### Molecular biology and transgenic strain generation

All constructs were cloned using the Multisite Gateway system (Invitrogen, Waltham, MA, USA) into pCG150 (*Merritt and Seydoux, 2010*). All constructs obtained from LR reactions were sequenced for verification. Plasmids and Fosmid that were used are listed in *Supplementary file 4*. The *tfap2-beta* gene was codon-optimized for *C.elegans* as described(*Redemann et al., 2011*).

### Transformation

We generated transgenic strains by microparticle bombardment or by microinjection using *unc-119 (ed3)* rescue as a selection marker (*Wilm et al., 1999*; *Praitis et al., 2001*). For fosmid isolation, we used the FosmidMAX DNA Purification kit (epicentre) or Qiagen plasmid midi kit.

### C10C6.7 mutant generation

The deletion alleles *C10C6.7(goe3)* and *C10C6.7(goe5)* were created with a CRISPR/Cas9 system as it was described before (*Friedland et al., 2013*). Target sequence for the sgRNA was GTTATGG TGAGAAGGAAAGCtgg. *The C10C6.7* gene locus was sequenced and the deletions were mapped to the second exon. They are 25 bp and 4 bp long, respectively, and cause a frame shift, thus most likely are molecular null alleles.

### Imaging in agarose microchambers

All long-term imaging experiments were carried out using agarose microchamber imaging as described (*Bringmann, 2011*; *Turek et al., 2015*).

For behavioral analysis, worms were filmed in a burst mode every 10–15 min for 20 s with a frame rate of 2 pictures/second. Nose tracking was performed manually. Mean velocities of nose speed were calculated for sleep and wake, where sleep was defined as the non-pumping phase and wake was defined as a 2-hr period directly before sleep.

Calcium imaging was performed similar as described before using GCaMP3.35 and co-expression of mKate2 as an expression control (*Schwarz et al., 2011*; *2012*; *Turek et al., 2013*; *2015*; *Schwarz and Bringmann, 2013*). For calcium imaging, we used an Andor (UK) iXon (512 x 512 pixels) EMCCD camera and LED illumination (CoolLed, UK) using standard GFP and Texas Red filter sets (Chroma, Bellow Falls, VT). Exposure times were in the range of 5-20 ms and allowed imaging of moving worms without blurring. The EMCCD camera triggered the LED through a TTL 'fire' signal to illuminate only during exposure. LED intensity was in the range of 15–30%. EM gain was between 50 and 250. All calcium-imaging experiments were done in agarose microchambers. Typically, 4–15 individuals were cultured in individual microchambers that were in close vicinity. Animals were filmed by taking a z-stack every 6 or 10 min or in a continuous mode, which means using a frame rate of 1 picture / 4 s. If more than four animals were filmed in parallel, individual compartments were repeatedly visited by using an automatic stage (Prior Proscan2/3, Rockland, MA) set to low acceleration speeds. Before each fluorescent measurement, we took a brief DIC movie to assess the developmental stage and behavioral state. Larvae that showed pharyngeal pumping were scored as being in the wake-like state. Movies were analyzed using homemade Matlab routines.

## Spinning disc imaging

For fluorescence imaging of reporter lines (*Figure 1C,D*, *Figure 2B*, *Figure 4C,D*, *Figure 1—figure supplements 1A,B*, *2B*, *3A,B*, *4A,B*, *5*), we used spinning disc imaging with an Andor Revolution spinning disc system using a 488 nm laser and a 565 nm laser, a Yokogawa (Japan) X1 spinning disc head, a 100x oil objective and an iXon EMCCD camera. Z stacks were taken and a maximum intensity projection calculated using iQ software.

## Optogenetics

Channelrhodopsin experiments were performed inside agarose microchambers as described (*Turek et al., 2013*). We grew hermaphrodite mother worms on medium that was supplemented with 0.2 mM all trans Retinal (Sigma-Aldrich, St. Louis, MO). We then placed eggs from these mothers together with food into microchambers without any further retinal supplementation. We stimulated Channelrhodopsin with an LED of 490 nm with about 0.36 mW/mm$^2$ as measured with a light voltmeter. Images were captured with an Andor Neo sCMOS camera (2560 x 2160 pixels).

Worms were filmed every 30 min for 60 s with a frame rate of two pictures / second. Channelrhodopsin stimulation with constant blue light was applied for 20 s starting after 20 s. Nose tracking was performed manually. We calculated mean velocities for wake using a period of 2 hr directly before sleep.

## Neuron identification

We crossed *goeIs290* and *goeIs285* in the following strains to identify neurons:

MU1085 (*Wightman et al., 2005*, EG1285 (*McIntire et al., 1997*), BZ555 (*Nass et al., 2002*), HBR1213, OH1422 (*Tsalik et al., 2003*), HBR887, HBR777, QW122 (*Donnelly et al., 2013*).

## Transcriptional profiling

For both wild-type and mutant conditions, four biological samples were collected. For transcriptional profiling of pretzel-stage embryos, each sample contained approximately 3000 animals that were picked manually into one ml of Trizol (Invitrogen). For transcriptional profiling of sleeping L4 larvae, each sample contained approximately 200 animals that were picked manually into one ml of Trizol (Invitrogen). Transcriptional profiling and microarray data analysis was done the same way as it was described before (*Turek and Bringmann, 2014*) except that fold change threshold was 1.5 and the GO term analysis was omitted. Microarray data was deposited at the GEO database and can be accessed using the following links:

http://www.ncbi.nlm.nih.gov/geo/query/acc.cgi?acc=GSE73282

http://www.ncbi.nlm.nih.gov/geo/query/acc.cgi?acc=GSE73283

## GSE73282, Transcriptional profiling of *C. elegans aptf-1(gk794)* mutant in L4 sleep stage

GSM1890106 aptf-1_1
GSM1890107 aptf-1_2
GSM1890108 aptf-1_3
GSM1890109 aptf-1_4
GSM1890110 N2_1
GSM1890111 N2_2
GSM1890112 N2_3
GSM1890113 N2_4

## GSE73283, Transcriptional profiling of *C. elegans aptf-1(gk794)* mutant in pretzel-stage embryos

GSM1890140 aptf-1_1 in pretzel-stage embryos
GSM1890141 aptf-1_2 in pretzel-stage embryos
GSM1890142 aptf-1_3 in pretzel-stage embryos
GSM1890143 aptf-1_4 in pretzel-stage embryos
GSM1890144 N2_1 in pretzel-stage embryos
GSM1890145 N2_2 in pretzel-stage embryos

GSM1890146 N2_3 in pretzel-stage embryos
GSM1890147 N2_4 in pretzel-stage embryos

## Heat-shock-based overexpression

For heat-shock-induced overexpression of *flp* neuropeptides, we cultured adult worms on NGM plates seeded with *E. coli* OP50 and sealed with parafilm. Heat shock was applied using a water bath at 37°C where the plates were placed for 5 min, the agar side facing the water. Worms were scored for moving / pumping behavior directly after heat shock and consecutively in time intervals of 30 min for 4 hr.

## Statistics

Statistical tests used were Wilcoxon Signed Paired Ranks test, Student's t-test or Welch test using Origin software. Error bars are SEM. For statistical analysis of overexpression experiments, a Fisher's exact test was done in Matlab.

## Acknowledgements

We thank the CGC, which is funded by NIH Office of Research Infrastructure Programs (P40 OD010440), the MITANI Lab through the National Bio-Resource Project of the MEXT(Japan), Yishi Jin, Mark Alkema, Mario de Bono, Victoria Butler, Bill Schafer, Chris Li, Jan Konietzka for strains and Mihail Sarov for fosmids, Ines Lewandrowski for injecting CRISPR/Cas9 constructs, the Transcriptome Analysis Laboratory Goettingen (headed by Gabriela Salinas-Riester) for microarray experiments and for data processing, Mascha Friedrich and Luisa Welp for assistance with manual data processing.

## Additional information

### Funding

| Funder | Grant reference number | Author |
| --- | --- | --- |
| Max-Planck-Gesellschaft | Max Planck Research Group | Henrik Bringmann |

The funders had no role in study design, data collection and interpretation, or the decision to submit the work for publication.

### Author contributions

MT, JB, Conception and design, Acquisition of data, Analysis and interpretation of data, Drafting or revising the article; J-PS, Conception and design, Acquisition of data, Analysis and interpretation of data; SK, Generated plasmids and strains, Contributed unpublished essential data or reagents; HB, Conception and design, Analysis and interpretation of data, Drafting or revising the article

### Author ORCIDs

Henrik Bringmann, http://orcid.org/0000-0002-7689-8617

## Additional files

### Supplementary files

• Supplementary file 1. Genes that are differentially expressed in *aptf-1(-)*. Table 1A, Genes with altered expression in *aptf-1(gk794)* mutant in pretzel-stage embryos. Table 1B, Genes with altered expression in *aptf-1(gk794)* mutant during L4 larvae sleep.

• Supplementary file 2. *C. elegans* strains. A list of *C. elegans* strains that were used for this study.

• Supplementary file 3. Primers. A list of primers that were used for this study.

• Supplementary file 4. DNA constructs. A list of DNA constructs (Plasmids and Fosmids) that were used for this study.

## Major datasets

The following datasets were generated:

| Author(s) | Year | Dataset title | Dataset URL | Database, license, and accessibility information |
|---|---|---|---|---|
| Turek M, Bringmann H, Salinas-Riester G | 2015 | Transcriptional profiling of *C. elegans aptf-1(gk7954)* mutant in L4 sleep stage | http://www.ncbi.nlm.nih.gov/geo/query/acc.cgi?acc=GSE73282 | Publicly available at the NCBI Gene Expression Omnibus (Accession no: GSE73282). |
| Turek M, Bringmann H, Salinas-Riester G | 2015 | Transcriptional profiling of *C. elegans aptf-1(gk7954)* mutant in pretzel-stage embryos | http://www.ncbi.nlm.nih.gov/geo/query/acc.cgi?acc=GSE73283 | Publicly available at the NCBI Gene Expression Omnibus (Accession no: GSE73283). |

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
