## [Decision Letter]

Thank you for submitting your work entitled "Sleep-active neuron specification and sleep induction require FLP-11 neuropeptides to systemically induce sleep" for consideration by *eLife*. Your article has been reviewed by three peer reviewers including Leslie Griffith, a Reviewing Editor and overseen by a Senior Editor.

The reviewers have discussed the reviews with one another and the Reviewing Editor has drafted this decision to help you prepare a revised submission.

Summary:

This is a very interesting paper that makes a good case for the ability of a neuromodulator to effect global state changes on the brain. For the most part, the data are tight and convincing. The study also provides novel insights into how sleep may be regulated in *C. elegans*. Particularly of note is the depth in which they have examined the molecular basis of neuronal differentiation to the functional state and the regulation of neuropeptide expression. These findings could potentially be extrapolated to understand the molecular underpinnings of sleep in other organisms.

Essential revisions:

1) The most problematic aspect of this study is the claim that FRPR-3/NPR-22/NPR-4 are cognate receptors for FLP-11. These claims are based on in vitro binding assays indicating effective concentrations for FLP-11 activation that are likely unphysiologically high. For example, for FRPR-3 the EC_50_ is around 1μM, and for NPR-22, the EC_50_ was too high to calculate, so they had to resort to stating the activity threshold (0.75μM-2.5μM). Additionally, for NPR-4, the EC_50_ was greater than 10μM. Given that most neuropeptides probably operate within the sub-nanomolar range, it is unlikely that these receptors are cognate for FLP-11.

One alternative model is that FLP-11 does not strongly activate these receptors in vivo to an appreciable level, but instead, activates other receptor(s) which may then (through x number of relays) trigger the release of additional ligands that bind to FRPR-3/NPR-22/NPR-4 for sleep. This is reflected by the mild sleep phenotype in the triple receptor mutant (Figure 4) when compared to the *flp-11* mutant (Figure 3). For the authors to claim that FLP-11 binds to these receptors, they would need to demonstrate receptor activation at physiological levels independently. Alternatively, the authors may wish to reconsider their conclusion and state that they have found neuropeptide receptors that partially suppress the over-expression of FLP-11, and loss of these receptors also lead to sleep defects while speculating on potential mechanisms.

2) There are also problems with the OE studies. It is known that heat shock of *C. elegans* can initiate quiescent states, which is problematic when testing for heat shock induced protein synthesis that may lead to sleep. An additional caveat is that the neuropeptide FLP-11 would be expressed globally throughout the animal. This unphysiological presence of a single neuropeptide could overwhelm the homeostasis of the nervous system and thus could cause non-specific quiescence. To exclude this possibility, the authors should try and test other peptides under the heat-shock promoter to demonstrate that the phenotype is specific to FLP-11 overexpression and is not caused by overexpression of any peptide. The authors should also describe in detail how the heat shock experiments were performed in the "Experimental Procedures" sections.

3) The language describing rescue of *aptf-1* with *flp-11* is overblown. It is *not* a complete rescue and saying that immobility was "restored" (subsection “RIS induces sleep through FLP-11”) is incorrect. As is the statement in the Figure 3 legend "rescue" should be "partial rescue". In fact, the rescued animals only show immobility that is at the level of the *flp-11* mutant. This strongly suggests that *flp-11* is not the only relevant downstream gene of *aptf-1*. This needs to be explicitly acknowledged.

---

## [Author Response]

We have rewritten the manuscript so that we describe the phenotypes that we found for the three GPCRs. We clearly state that it is unclear whether these receptors bind to FLP-11 in vivo and that additional receptors need to be identified.

We included control animals that did not express the transgene to test for heat shock-induced quiescence in our initial experiment. We kept the heat shock short (5 min). Under these conditions the control worms did not show quiescence. We changed the color of this control data in the panel so that it can be found more easily. We also added a paragraph to the Methods section that describes this experiment. We tested OE of three additional peptides and found that they did not induce quiescence, suggesting that quiescence cannot be induced by any peptide.

We have made clear in the text that the rescue is partial. We have added a sentence to the paragraph that states that *aptf-1* likely acts through other genes in addition to *flp-11*.